# Lifetime Costs of Surviving Cancer—A Queensland Study (COS-Q): Protocol of a Large Healthcare Data Linkage Study

**DOI:** 10.3390/ijerph17082831

**Published:** 2020-04-20

**Authors:** Katharina M. D. Merollini, Louisa G. Gordon, Joanne F. Aitken, Michael G. Kimlin

**Affiliations:** 1Sunshine Coast Health Institute, School of Health and Sport Sciences, University of the Sunshine Coast, Maroochydore, QLD 4558, Australia; michaelkimlin45@gmail.com; 2QIMR Berghofer, Medical Research Institute, Herston, QLD 4006, Australia; Louisa.Gordon@qimrberghofer.edu.au; 3School of Nursing, Queensland University of Technology, Kelvin Grove, QLD 4059, Australia; 4School of Public Health, The University of Queensland, Herston, QLD 4006, Australia; 5Cancer Council Queensland, Fortitude Valley, QLD 4006, Australia; JoanneAitken@cancerqld.org.au; 6Institute for Resilient Regions, University of Southern Queensland, Ipswich, QLD 4305, Australia; 7Menzies Health Institute Queensland, Griffith University, Gold Coast, QLD 4222, Australia; 8School of Biomedical Sciences, Queensland University of Technology, St Lucia, QLD 4072, Australia

**Keywords:** cancer survivors, health service use, costs and cost analysis, long-term outcomes, economic models, health economics

## Abstract

Australia-wide, there are currently more than one million cancer survivors. There are over 32 million world-wide. A trend of increasing cancer incidence, medical innovations and extended survival places growing pressure on healthcare systems to manage the ongoing and late effects of cancer treatment. There are no published studies of the long-term health service use and cost of cancer survivorship on a population basis in Australia. All residents of the state of Queensland, Australia, diagnosed with a first primary malignancy from 1997–2015 formed the cohort of interest. State and national healthcare databases are linked with cancer registry records to capture all health service utilization and healthcare costs for 20 years (or death, if this occurs first), starting from the date of cancer diagnosis, including hospital admissions, emergency presentations, healthcare costing data, Medicare services and pharmaceuticals. Data analyses include regression and economic modeling. We capture the whole journey of health service contact and estimate long-term costs of all cancer patients diagnosed and treated in Queensland by linking routinely collected state and national healthcare data. Our results may improve the understanding of lifetime health effects faced by cancer survivors and estimate related healthcare costs. Research outcomes may inform policy and facilitate future planning for the allocation of healthcare resources according to the burden of disease.

## 1. Introduction

Globally, the number of cancer survivors has exceeded 32 million [1] with an estimated 16.9 million people currently affected in the United States (US) alone [2] and over one million in Australia [3]. The most widely used definition of cancer survivorship is from the National Coalition for Cancer Survivorship in the US and includes for each person the period ‘from the time of diagnosis, through the balance of his or her life, regardless of the ultimate cause of death’ [4,5]. Different stages of survivorship comprise acute (diagnosis to treatment), chronic (ongoing) and long-term/late survivorship (≥5 years post diagnosis). One in two Australians will be diagnosed with cancer during their lifetime and cancer is now the second most common cause of premature death. Over 144,000 new cancer diagnoses were predicted for Australia in 2019 (excluding squamous and basal cell carcinomas) and this number is expected to increase each year as the population ages [6]. Over time survival of cancer has improved significantly and overall 5-year survival for invasive cancers is now 68% [6]. The trend of increasing cancer incidence and extended survival has resulted in increasing numbers of cancer survivors and, potentially, an increasing burden on healthcare systems [7,8,9,10]. The latest figures released by the Australian Institute of Health and Welfare (AIHW) stated $4526 million Australian dollars (AUD) healthcare expenditure on cancer management per year (for 2008/2009) and represent a 56% increase within only 8 years [11]. These costs only captured direct treatment for cancer and have not officially been updated since. The financial costs of cancer are enormous, both for the patients themselves and for the health system, but these costs extend far beyond the initial diagnosis and cancer treatment. It is estimated that more than 50% of cancer survivors suffer late effects, such as physical (pulmonary, cognitive and cardiac effects, subsequent malignant neoplasms) and psychosocial effects (depression, anxiety and fear of recurrence) [12,13,14] that are likely to need ongoing healthcare. Childhood cancer patients face additional health risks due to exposure to radiation and chemotherapy during early life growth and development. Recent literature reviews reported the following late adverse effects childhood cancer: increased risk of subsequent primary malignancies, hypothyroidism, severe fatigue, hearing loss organ dysfunctions (ovarian, liver, lungs, heart, kidney and bladder), obesity as well as musculoskeletal, dental and psychosocial effects [15,16,17,18,19,20,21].

The Australian healthcare system consists of both public and private providers. All services and treatments in public hospitals are provided at no cost to Australian citizens and permanent residents [22]. Medicare is Australia’s universal health care system and it covers many in-hospital and out-of-hospital services delivered privately as well as pharmaceuticals. Pharmaceuticals are subsidized through the Pharmaceutical Benefit Scheme (PBS) and a health safety net for low-income earners is available with discounts on co-payments up to a certain annual threshold. Private health providers such as community general practitioners (GPs), private hospitals, specialists and other allied services operating privately, are free to set their fees individually without government regulation. Individuals can choose to have private health insurance if they wish to be covered for services and treatments in private hospitals, but insurers do not cover GP services, diagnostics and pathology services. Patients must cover the gap between government subsidies and private fees charged, in out-of-pocket or co-payments. Cancer patients typically require access to providers from both public and private health care over a prolonged period of time and need to cover a portion of their treatment expenses. A significant number of cancer survivors suffer financial hardship as a consequence of cancer treatment, previously described as ‘financial toxicity’ [23,24]. Out-of-pocket costs are a concern for cancer patients and recent Australian studies have described these important costs for indigenous vs non-indigenous cancer patients [25], Queenslanders with major cancer [26] and rural Western Australian cancer patients [27]. Other jurisdictions have also noted the financial hardship and high out-of-pocket expenditure due to cancer, such as the US, Germany and Canada [28,29,30]. The lifetime burden for pediatric cancer sufferers and their families is expected to be particularly high but little is known about this problem in Australia and previous studies were based on small samples [31,32,33,34,35,36].

To date no other Australian study has estimated long-term health care costs of cancer patients on a population level. Three recent studies estimated short to medium-term costs of cancer. Goldsbury et al. [37] reported results of patients in New South Wales (NSW) aged 45 and up from the year before diagnosis to 5 years after (2009–2013). Based on a sample of 266,000 people, they estimated AUD $6.3 billion for the direct health services cost of cancer care in 2013 [37]. The results did not distinguish disease stages or age at diagnosis which are known to be major cost drivers [8]. Bates et al. [38] recently published total healthcare system costs in Queensland for all cancers during the first year post diagnosis (2011–2015) of AUD $4.3 billion and in addition patient co-payments of AUD $127 million. They also noted that patient co-payments were lower for indigenous Australians and people in regional areas but higher for people from least disadvantaged socioeconomic backgrounds but did not look at long-term outcomes [38]. Callander et al. [25] estimated out-of-pocket costs for people with cancer for the first three years after diagnosis and found that indigenous people had less than half the expenditure and accessed fewer Medicare services compared to non-indigenous patients. 

To our knowledge no other study has investigated long-term health and cost outcomes of all cancer survivors on a population level. More specific information is needed on the scope of ongoing healthcare and health service delivery needs and cost due to previous cancer diagnosis in the Australian context. Economic modeling provides an ideal platform to predict future healthcare expenditure and the potential impact of more investment in cancer prevention. The aims of this project are:(1)To quantify lifetime health service use by cancer patients from the date of diagnosis.(2)To identify factors associated with high and low health service use.(3)To estimate lifetime health service use and cost by cancer type from the date of diagnosis using economic modeling.(4)To identify opportunities for improvement for future health service delivery.

## 2. Materials and Methods

### 2.1. Study Cohort

Every Queensland resident diagnosed with a first primary cancer, excluding basal and squamous cell carcinoma, between January 1997 and December 2015 will be eligible, for the study, giving a total estimated sample size of 475,000 people. Individuals will be identified from the Queensland Cancer Register (QCR), a dataset containing records of all Queensland residents diagnosed with cancer. Cancer, except for basal and squamous cell carcinoma, is a legally notifiable condition in Queensland. Individuals experiencing a relapse of cancer will not be defined as new cases.

### 2.2. Ethical Approval and Consent to Participate

Ethics approval was obtained from the University of the Sunshine Coast Human Research Ethics Committee (USC HREC Approval A/17/941) and from the Australian Institute of Health and Welfare (AIHW) Human Research Ethics Committee (EO2017/3/348). Approval for Queensland data extraction and linkage, including a waiver of consent was sought from Queensland Health under the Public Health Act from 2005 and was granted by the Director General (grant RD007281).

The waiver of consent was justified as there will not be any direct participation or interaction between participants and researchers; Researchers do not wish to identify individual participants and data analysis will be conducted using de-identified data. Personal information will only be visible by the Queensland data custodians and AIHW linkage officers but not passed on to researchers. Information collected will be used for data analyses as described in the methods section below. No individual records will be published, all publications will include aggregated data analyses and results will be reported in summarized form.

### 2.3. Data Collection 

The following six administrative databases containing routinely collected data will be linked using name, date of birth and other variables listed in detail in Appendix A:

Queensland data:*(1)* Queensland Cancer Register (QCR)*(2)* Queensland Hospital Admitted Patient Data Collection (QHAPDC)*(3)* Emergency Department Information System (EDIS)*(4)* Healthcare Purchasing & System Performance data (HPSP), HHS Funding Models: National Hospital Cost Data Collection (NHCDC), Patient Activity Weighting (PAWS/ GenWAU)

National data:*(5)* Medicare Benefits Schedule (MBS)*(6)* Pharmaceutical Benefit Schedule (PBS)

### 2.4. Data Items Collected

Details of each database are described in Appendix A, including purpose for collection, data extraction timeframe, variables required for data linkage (linkage variables) and variables required for data analyses (research variables).

### 2.5. Data Linkage Process

The linkage process is summarized in Figure 1. As a first step the Queensland Cancer Register (QCR) establishes the initial patient cohort. These data are passed on to the Statistical Services Branch (SSB) at Queensland Health to link the other Queensland (Qld) data sets as following:SSB formats QCR cohort as required by AIHW. All identifying information to be included to allow AIHW to identify cases. A new, random ID is attached to each case patient, e.g., ‘CASE00000001’.QHAPDC & EDIS data are linked by SSB and episode IDs of all patient cohort records are transferred to HPSP.Extracted data by HPSP will be transferred back to SSB for matching back to linked records.SSB will complete extraction of all Queensland data (including QCR data), and transfer this to a Secure Unified Research Environment (SURE) administered by the Sax Institute in Sydney, Australia.SSB transfers cohort file to AIHW using secure file transfer service.

The AIHW has authority to extract and link Commonwealth data and to perform linkage of state and national data. The AIHW will receive Queensland data and link this to appropriate MBS and PBS records. The final dataset will be transferred from AIHW and SSB linkage units to SURE. The research team will be notified and will receive a key for remote data access.

### 2.6. Data Storage

The linked, de-identified records will be held in a secure, password-protected database on SURE servers with a regular backup routine. Only the approved researchers stated will have access to the data and will have remote access to the secure file after completion of a training module for SURE users. The linked data will be kept for a total of 10 years and will be destroyed/ deleted by SURE in January 2028. This is in accordance with the minimum retention time of data under the Australian Code for the Responsible Conduct of Research requiring data storage until 5 years after completion of publication of results.

### 2.7. Analyses

#### 2.7.1. For Aims 1–2: Analysis of Linked Healthcare Records Over Time

For each person in the cohort, the linked data will capture most of healthcare usage (including GP appointments, public and private hospitalizations, emergency treatments, allied health professional utilization) and provide information on pharmaceuticals and medical conditions.

Aim (1) Quantify health service usage and associated costs:

Initially, all datasets will be checked for completeness, logical errors and missing data. Linked datasets from state and national data linkage teams will be merged and the consistency of linkage variables checked (e.g., selecting patient IDs at random and comparing related demographic information across different datasets). If discrepancies are found, we will send these to the data linkage teams for possible correction. If that is not possible simple data imputation will be used for minor missing values and if necessary incomplete cases will be excluded from further analyses.

We will use a bottom-up costing approach which gives the most accurate results in determining health service use and allows for total cost calculations per patient and subgroup [39]. Overall health service utilization will be calculated per year to compare overall cost differences between different age groups and types of services utilized (MBS broad type of service use, MBS item category). Overall costs are defined as the product from each individual cost component, consisting of pharmaceuticals, medical & allied health services, hospital admissions, emergency presentations and healthcare purchasing data (see costing model in Figure 2 below). Essential for deriving costs from medical records will be International Classification of Diseases, 10^th^ Revision (ICD-10) codes (hospital, emergency), MBS data on item description, number of services, fees charged, schedule fee and benefits paid, as well as PBS data on drug description (item number, name, Anatomical Therapeutic Code, form & strength), number of scripts, patient contribution and net benefit paid. Direct cost of hospitalization will be calculated by utilizing QHAPDC records on patient admissions, cost for length of stay and ICD-10 codes; the cost of all other private and community healthcare services provided and patient co-payments will be derived via Medicare claims records. The overall health service utilization and distribution of associated costs for all Queensland cancer patients included in this study will be detailed by type of cancer, phase of care, time since diagnosis, age and gender.

Aim (2) Identify factors associated with high/low health service usage/cost:

We will develop a detailed data analytical plan to undertake statistical analyses of cost data and identify factors associated with high/low health service usage/cost, such as socio-demographic (e.g., geographic location, ethnicity, age, sex) and clinical characteristics (e.g., type of cancer, time since diagnosis). We will consider appropriate statistical approaches, such as non-parametric bootstrapping [40], cluster analysis [41], quantile regression analyses [42], two-part models (if zero values are an issue), quintile regression (splitting cost outcomes into 5 levels) and Generalized Linear Models (GLM) [43]. The latter are particularly suitable for highly right-skewed data, such as health care costs. The results will inform decision-makers and can guide health policy.

#### 2.7.2. For aim 3: Economic Modeling Estimating Lifetime Outcomes of Surviving Cancer (Based on Findings from Aims 1–2)

Model overview: The results from Aim 1 will be used to estimate lifetime costs of surviving cancer by cancer type using appropriate economic models (such as Markov models / decision-analytic models). Different cost-effectiveness analyses will be conducted, based on observed costs and effects of survivors with different profiles (type of cancer, risk group, treatment pathway or remoteness category) to highlight where the greatest gains to health and costs are likely to be made. Long-term cost and health outcomes of individual cancer patients will be incorporated into economic models to estimate overall societal outcomes and predict future healthcare usage faced by the Australian healthcare system. Time series analyses will be used to enable prediction of future trends.

Model input parameters & perspective: Applying cohort simulation techniques, the model will capture health and cost outcomes of patients moving through relevant model health states during their life (disease onset until death). If necessary, results beyond the scope of the administrative data will be extrapolated to the end of the model using appropriate parameter estimation methods [44]. Results from Aim 1 will provide estimates of health service utilization and the proportion of patients’ co-payments. Health outcomes will be measured as life years (as observed in the data) and/or quality-adjusted life year (QALY) estimates from the literature. Model pathways (transition probabilities) will be informed by the actual order of events in the data and the modeling software TreeAge Pro will be used for the analyses. Relevant epidemiological data will be included, such as underlying mortality by gender and age. The model simulation will end when all patients in the hypothetical cohort are in a death state (terminal state). The models will adopt a health services perspective and were possible, a societal perspective. The latter will also include costs beyond the scope of routinely collected data, such as productivity losses (presenteeism, absenteeism, premature mortality) patient out-of-pocket costs and career time. We will adhere to best practice guidelines for economic evaluations and modeling of uncertainty [45,46]. Future costs and effects will be discounted at 3.5%. 

Model evaluation: The results will be presented as Incremental Cost-Effectiveness Ratio (ICER) whereby incremental costs ∆C are divided by incremental effects ∆E. We will also calculate and Net Monetary Benefits (NMB) whereby NMB = (λ*∆E)-∆C [47]. Societal willingness-to-pay (WTP = λ) is the valuation of health in monetary terms, i.e., the willingness-to-pay threshold for one extra unit of benefit. Uncertainty will be explored in deterministic sensitivity analyses where model input parameters are varied by inserting a range of possible values and effect on the model is observed. Examples are 1-way or 2-way sensitivity analyses and graphic illustration of parameters with greatest influence as Tornado diagrams. Furthermore, probabilistic sensitivity analyses (PSA) will be performed, where distributions are fitted around model input parameters and values from each distribution are drawn at random at the same time. Model uncertainly will be plotted in Cost–Effectiveness–Acceptability Curves showing ICER for different willingness-to-pay thresholds. PSA results will also be illustrated as clouds on the cost-effectiveness plane [48]. Sensitivity analyses will also allow for changes in health policy and subsequent changes in cancer prevalence to be reflected in the model; different plausible changes will be included and the effect on overall model outcomes tested.

#### 2.7.3. For Aims 4: Identifying Opportunities of Improvement for Future Health Service Delivery

Overall results will be used to advise policy makers on the expected future projection of healthcare resource use and service delivery needs. We will also look at potential areas of improvement, such as the impact of effective prevention strategies on long-term outcomes (e.g., hypothetical reduction in risk factors), particular high cost services and improvements in clinical processes likely to result in cost-savings (e.g., guideline adherence). For example, potential sources of system inefficiencies can be identified by analyzing the level of adherence to treatment guidelines, including prescribed drugs (PBS data), medical services (MBS data) and hospital treatment (QHAPDC data) received. Similarly, we will identify and quantify potential instances of ‘low value’ care where no net patient benefit is expected but the service incurs unnecessary health system costs (e.g., sentinel lymph node biopsy for melanoma in situ) [49].

## 3. Discussion

This project will improve the understanding of long-term health consequences faced by cancer survivors. It will uncover major cost drivers, predict lifetime costs using economic modeling, reveal inefficiencies and quantify health service utilization on a population level in Queensland, Australia. By linking state and national healthcare data we will capture the whole journey of health service contact (incl. emergency presentations, GP and specialist appointments, public and private hospitalizations, pharmaceuticals, mental health services) and we will be able to estimate related costs of all cancer patients diagnosed and treated in Queensland over a period of up to 20 years.

A limitation of our study is that our retrospectively linked administrative healthcare data does not include indirect costs, such as productivity costs due to absenteeism, presenteeism and premature mortality, financial toxicity, career and family costs (e.g., travel, time of work). Nevertheless, we will try to include some of these major cost drivers in the predicted lifetime costs from a societal perspective using relevant estimates from the literature [50,51] and our own calculations. Furthermore, our budget did not allow for case-control matching to calculate excess costs of cancer, as shown by Goldsbury et al. [37] in their 45 and Up Study, but we will still be able to show overall cost increases and compare our estimates to their reported costs for cancer. Not having control cases will not affect our study aims and we will be able to report on overall healthcare costs incurred since cancer diagnosis. Another limitation of our data are that squamous and basal cell carcinomas are not recorded by the Queensland Cancer Register. Although these patients are hence excluded in our cohort of cancer patients with first primary malignancy, the healthcare costs of cancer patients also receiving biopsies or treatment for these common types of cancer (while having a different type of first primary cancer) will be captured in the overall healthcare costs. The current medical literature does not contain many long-term population-based studies on costs or the economic impact of cancer and the most comprehensive research in this context to date originated in Canada. In 2017 De Oliveira et al. compared lifetime costs (pre-diagnosis to death) of adult cancer patients in two large Canadian provinces based on linked data extracted over a ten-year period (1997–2007) [52]. Their results revealed that cancer care costs varied considerably by site, phase of care and time horizon. Another study published by De Oliveira et al. in 2018 [53] appraised the economic burden of cancer on a population level in the Canadian state of Ontario and found costs increased rapidly from $2.9 billion Canadian dollars (CA) in 2005 to CA $7.5 billion in 2012. Healthcare costs reported for children with cancer were higher than for adolescents or adults [54].

Other studies have attempted to estimate the costs of cancer on a population level but these were either for the short term [9,55,56] and/or focused on selected type/s of cancer [8,57,58,59,60,61,62,63,64]. For example, Luengo-Fernandez et al. [55] performed a population-based cost analysis for breast, colorectal, lung and prostate cancer across the European Union (27 countries) from a societal perspective, including healthcare costs, informal care costs and productivity losses. They estimated costs for the year 2009 at €126 billion Euro to the EU with direct health care costs making up about 40% of overall costs but also found large variance between countries which required further research. No long-term outcomes were considered. Sam et al. [56] compared healthcare costs for cancer vs common non-cancer morbidities and related population-level administrative data over a three-year period in Alberta, Canada. They found hematologic malignancies incurred the highest costs (CA $70 million), followed by colon cancer (CA $51 million) and lung cancer (CA $44 million). Again, no long-term costs were considered. Other recent research focused on long-term outcomes for a certain type of cancer, such as Corral et al. [60] for colorectal cancer treatment in Spain and Schernberg et al. [63] for head and neck cancer in France.

In summary, it is important to quantify healthcare costs relating to cancer in detail as cost parameters are a crucial input parameter for economic evaluations and are used for decisions regarding resource allocation [65]. We will contribute to the growing literature on long-term healthcare costs incurred in Australia since cancer diagnosis and provide detailed estimates that can be used for future economic evaluations by reporting healthcare costs (including measures of variance) according to type of cancer, age, sex, phase of care or time since diagnosis. This will provide valuable resources for future cost-effectiveness evaluations in the broader field of cancer, including evaluations of screening, treatment and prevention. Our research outcomes will also form a body of evidence to show potential lifetime cost savings by cancer prevention. We will inform policy makers in Queensland/ Australia and hence facilitate future planning on the utilization and allocation of healthcare resources according to the burden of disease and potentially lead to more investments in cancer prevention and survivorship care.

## 4. Conclusions

Long-term outcomes and health system utilization after cancer diagnosis are currently not fully understood. Our project will investigate long-term costs following cancer diagnosis, cost drivers and opportunities for improving efficiencies in the current system using administrative population level data of up to 20 years. Projecting and accurately planning for the future burden of disease of cancer survivors will allow for adequate resource allocation by policy makers, potentially more investment in cancer prevention activities and result in the health system running more efficiently to the benefit of the wider community. More research is warranted to better understand the ever-growing cohort of cancer survivors and their long-term outcomes on a population level.

## Figures and Tables

**Figure 1 ijerph-17-02831-f001:**
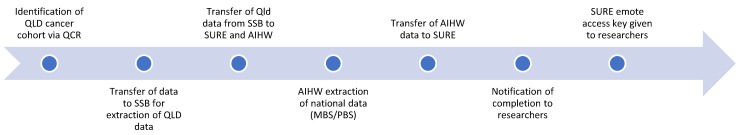
Data linkage process.

**Figure 2 ijerph-17-02831-f002:**
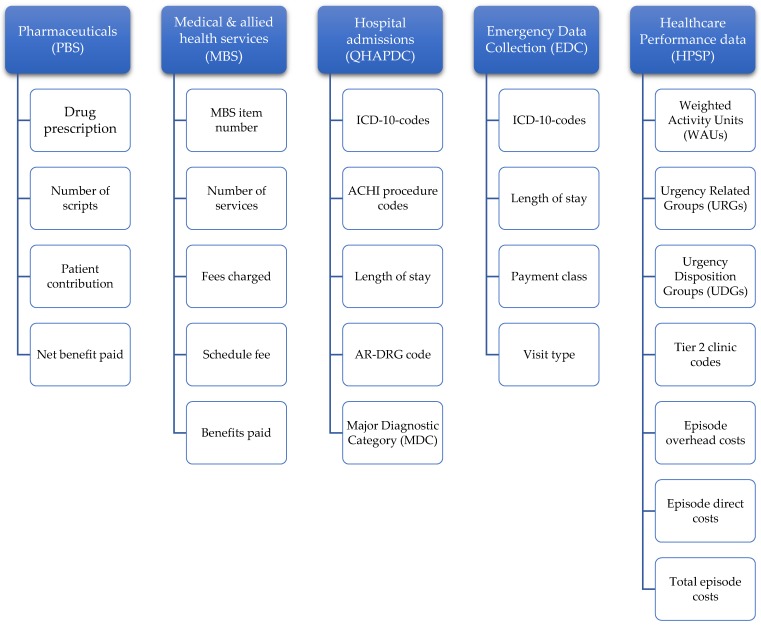
Individual cost components part of bottom-up costing model *. * PBS = Pharmaceutical Benefit Scheme; MBS = Medical Benefit Scheme; QHAPDC = Queensland Hospital Admitted Patient Data Collection; HPSP = Healthcare Purchasing and System Performance Data; ICD-10 = International Classification of Diseases, 10^th^ Revision; ACHI = Australian Classification of Health Interventions; AR-DRG = Australian Refined Diagnosis Related Groups;.

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
