# Peer review of "Lifetime Costs of Surviving Cancer—A Queensland Study (COS-Q): Protocol of a Large Healthcare Data Linkage Study"

_ijerph, 2020, doi:10.3390/ijerph17082831_

Round 1

Reviewer 1 Report

I believe this is an excellent proposal and protocol for use. There are several writing corrections needed but otherwise has the potential for an excellent study.

One comment is that it would have been more scholarly if the data analysis methods were presented.

Author Response

Thank you very much for your excellent feedback.

As suggested, we added substantially more detail to the materials and methods section (line 120)in particular to 2.1 Study Cohort, 2.2 Ethical Approval & Consent to Participate and to 2.7 Analyses (line 196).  

Reviewer 2 Report

The study entitled “Lifetime costs of surviving cancer – a Queensland 2 study (COS-Q): protocol of a large healthcare data 3 linkage study.” The results will improve understanding of lifetime health effects faced by cancer survivors and estimate related healthcare costs.

The paper is well written and organized, but its needs minor changes before publishing. Minor issues that authors should be addressed before the manuscript will be considered again for publication.

Introduction

- The introduction section is quite complete and contains theoretical background. However, more references are needed especially in the statements where the citations are missed.  For example, in lines 44 y 45  “Globally, the number of cancer survivors has exceeded 32 million with an estimated 16.9 million 44 people currently affected in the United States (US) alone and over 1 million in Australia [1].” It is recommended includes the reference of Australian data.

Material and Methods

- It is important to find out if relapses are going to consider a new case or not.

- In the Figure 2, it is necessary includes the significant of acronym again.

- Ethical approval and consent to participate. It is recommendable amplify the information about waiver of inform consent.

Limitations

- The squamous and basal cell carcinomas are excluded in the registers. Even, I can understand why, authors should comment their reason to exclude these frequent cancer in general population and if it affects the potential results. I suggest that authors comment as a limitation of the study or in discussion section.

Author Response

Thank you very much for your excellent feedback. We added substantially more detail to the methods / analyses section. Furthermore, please see our responses below. 

Introduction 

The introduction section is quite complete and contains theoretical background. However, more references are needed especially in the statements where the citations are missed.  For example, in lines 44 y 45  “Globally, the number of cancer survivors has exceeded 32 million with an estimated 16.9 million 44 people currently affected in the United States (US) alone and over 1 million in Australia [1].” It is recommended includes the reference of Australian data. 

We would like to thank the reviewer for the opportunity to amend this oversight. We added the following references: 

  • Rowland, J.; Yabroff, R. Cancer Survivorship. In The Cancer Atlas, Third Edition. ed.; Jemal A, T.L., Soerjomataram I, F, B., Eds. American Cancer Society: Atlanta, 2019; pp. 66-67. 
  • American Cancer Society. Cancer Treatment & Survivorship Facts & Figures 2019-2021; Atlanta, 2019. 
  • Cancer Council Victoria. Australians living with and beyond cancer in 2040. Availabe online: https://www.cancervic.org.au/research/registry-statistics/statistics-data/cancer-prevalence-in-2040.html (accessed on 05/04/2020). 

Material and Methods 

It is important to find out if relapses are going to consider a new case or not. 

We will follow patients with a first primary cancer diagnosis and will follow their disease progression. For patients with relapse that will not be considered a new case of cancer. To make this clear in the following statement was added to 2.1 Study cohort (line 118): 

‘Individuals experiencing a relapse of cancer will not be defined as new cases.’ 

In the Figure 2, it is necessary includes the significant of acronym again. 

The following footnote was added to Figure 2 outlining missing acronyms as a footnote for the Figure: 

* PBS = Pharmaceutical Benefit Scheme; MBS = Medical Benefit Scheme; QHAPDC = Queensland Hospital Admitted Patient Data Collection; HPSP = Healthcare Purchasing and System Performance Data; ACHI = Australian Classification of Health Interventions;  

Ethical approval and consent to participate. It is recommendable amplify the information about waiver of inform consent. 

The following paragraph was added to section ‘2.2 Ethical approval and consent to participate’ (l. 127): 

‘The waiver of consent was justified as there will not be any direct participation or interaction between participants and researchers; Researchers do not wish to identify individual participants and data analysis will be conducted using de-identified data. Personal information will only be visible by the Queensland data custodians and AIHW linkage officers but not passed on to researchers. Information collected will be used for data analyses as described in the methods section below. No individual records will be published, all publications will include aggregated data analyses and results will be reported in summarised form.’ 

Limitations 

The squamous and basal cell carcinomas are excluded in the registers. Even, I can understand why, authors should comment their reason to exclude these frequent cancer in general population and if it affects the potential results. I suggest that authors comment as a limitation of the study or in discussion section. 

The following paragraph was added as a limitation in section ‘4. Discussion’ (.279): 

‘Another limitation of our data is that squamous and basal cell carcinomas are not recorded by the Queensland Cancer Register. Although these patients are hence excluded in our cohort of cancer patients with first primary malignancy, the healthcare costs of cancer patients also receiving biopsies or treatment for these common types of cancer (whilst having a different type of first primary cancer) will be captured in the overall healthcare costs.’ 

Reviewer 3 Report

This paper studied the cost of cancer survival among Australians with the "Lifetime costs of surviving cancer-a Queensland study (COS-Q): protocol of a large healthcare data linkage study".
However, it is still in the early stages of collecting multiple databases, and the results are insufficient to suggest the direction for improvement in providing future medical services for long-term costs after cancer diagnosis.
In order to estimate the long-term cost for cancer patients, it would be a better paper to find out which factors are most influential and how to estimate them through a predictive model.
This paper also does not provide a clear method for estimating the cost of long-term cancer patients, and seems to be lacking in many areas.
Although it is said to be the basis for designing a long-term predictive model for cancer survivors, more specific studies need to be conducted in order to make long-term results at an increasing cohort and population level of cancer survivors more effective.

Author Response

Thank you for your excellent feedback. As noted by the reviewer, the study is in its early stages (data collection) and the results are unknown, hence the reviewer criteria of ‘are the results clearly presented’ and ‘are the conclusions supported by the results’ do not apply. 

However, one of the analysis aims (aim 4, l250) is to identify opportunities of improvement for future health service delivery. This means that we will try to identify areas for improvement or system inefficiencies (suggesting a positive direction of improvement) as described in section 2.7.3 (line 250) there is already existing literature indicating the direction of improvement. 

Reviewer comment: In order to estimate the long-term cost for cancer patients, it would be a better paper to find out which factors are most influential and how to estimate them through a predictive model. 

As described in the Analyses section aims 1-2 include to ‘Identify factors associated with high/low service usage/cost’ (line 197), hence factors influential on cost outcomes. We will then include these factors in our predictive economic models of lifetime outcomes, as indicated by the following heading in line 252: 

‘2.7.2 For aim 3: Economic modelling estimating life-time outcomes of surviving cancer (based on findings from aims 1-2)’ 

Reviewer comment: This paper also does not provide a clear method for estimating the cost of long-term cancer patients, and seems to be lacking in many areas. Although it is said to be the basis for designing a long-term predictive model for cancer survivors, more specific studies need to be conducted in order to make long-term results at an increasing cohort and population level of cancer survivors more effective. 

As a response to the reviewer’s concern we refined our methods section and added substantially more detail to section 2.7 Analyses (line 196). 

We believe that our methods are sufficient to build predictive health economic models using data from this study and if necessary, estimates from the literature to complete data inputs (which is commonly done in economic evaluations). Although our research might not provide answers to all possible questions, we will adhere to best practice guidelines for economic evaluations using the highest level of quality data available for each model parameter and potentially including high level evidence from other published literature. We will be clear about model assumptions and limitations of our data.  

Round 2

Reviewer 3 Report

A life-cycle approach is very important in assessing health and disease determinants.
Medical costs, or many of the factors in health, have been influencing many years, many life stages, or generations in a lifetime. In particular, in order to systematically analyze cancer, it is necessary to analyze the distribution of medical expenses according to age and gender, and to analyze how much cancer brings burden to an individual through life.
Therefore, it is necessary to estimate the medical cost of cancer for life.
In addition, the factors affecting cancer should be identified and used as basic data for establishing health policies.
Therefore, it is necessary to develop and build a model that is more specific and can reflect reality. In this paper, further analysis for systematic model development is still needed.
Epidemiological data, such as cancer-related health behaviors, prevalence, and mortality, are needed, and time-series data analysis is required to predict continued trends.
In the lifetime perspectives, it is necessary to analyze the financial effects of health care policies by considering a model for prevalence changes.
It will be a better paper if it becomes a paper that considers the contents mentioned above in this paper.

Author Response

The authors' reply to the report by reviewer 3 is shown in blue (italic) below. We would like to thank the reviewers for their time and feedback provided.

1) A life-cycle approach is very important in assessing health and disease determinants. Medical costs, or many of the factors in health, have been influencing many years, many life stages, or generations in a lifetime. In particular, in order to systematically analyze cancer, it is necessary to analyze the distribution of medical expenses according to age and gender, and to analyze how much cancer brings burden to an individual through life. Therefore, it is necessary to estimate the medical cost of cancer for life.

In accordance with the reviewer’s comments we added the following to the methods section.

  • In section 2.7.1 for Aim 1) ‘Quantify health service usage and associated costs’ (l200) the last sentence of this paragraph was amended to reflect that the analyses will include distributions of costs according to age and gender: ‘The overall health service utilisation and distribution of associated costs for all Queensland cancer patients included in this study will be detailed by type of cancer, phase of care, time since diagnosis, age and gender.’
  • Section 2.7.2 For aim 3 in ‘model overview’ (l257) the following was added to clarify the estimation of lifetime costs based on individual data: ‘Long-term cost and health outcomes of individual cancer patients will be incorporated into economic models to estimate overall societal outcomes and predict future healthcare usage faced by the Australian healthcare system.’

We would like to note that this project is not designed to estimate lifetime costs for each cancer patient. Our data collection started with cancer diagnosis for a maximum of 20 years (depending on year of diagnosis), until December 2016. We will be able to analyse data by age and gender but analyses of generational differences (>30 years) are beyond the scope of this project.

2) In addition, the factors affecting cancer should be identified and used as basic data for establishing health policies.

In the methods section 2.7.1 for Aim 2) Identify factors associated with high/low health service usage/cost’ (l 242) the following sentence was added at the end of the paragraph: ‘The results will inform decision-makers and can guide health policy.’

3) Therefore, it is necessary to develop and build a model that is more specific and can reflect reality. In this paper, further analysis for systematic model development is still needed. Epidemiological data, such as cancer-related health behaviors, prevalence, and mortality, are needed, and time-series data analysis is required to predict continued trends.

 ‘2.7.2 For aim 3: Economic modelling estimating life-time outcomes of surviving cancer’ in ‘model overview’ (l265) the following was added: ‘Time series analyses will be used to enable prediction of future trends.’ In model input parameters & perspective (l274) we added: ‘Relevant epidemiological data will be included, such as underlying mortality by gender and age.’

Although models can never be an exact representation of reality, we will include the most suitable model input parameters. Cancer-related health behaviour is internally reflected in health care usage patterns; therefore, no additional, external health behaviour data will be added to the model.

4) In the lifetime perspectives, it is necessary to analyze the financial effects of health care policies by considering a model for prevalence changes. It will be a better paper if it becomes a paper that considers the contents mentioned above in this paper.

In model evaluation (l295) the following was added: ‘Sensitivity analyses will also allow for changes in health policy and subsequent changes in cancer prevalence to be reflected in the model; different plausible changes will be included and the effect on overall model outcomes tested.’